# The effect of race/ethnicity on obesity traits in first year university students from Canada: The GENEiUS study

Tanmay Sharma[1]ⓔ, Baanu Manoharan[1]ⓔ, Christine Langlois[1], Rita E. Morassut[1], David Meyre[1,2]*

1 Department of Health Research Methods, Evidence, and Impact, McMaster University, Hamilton, Canada,
2 Department of Pathology and Molecular Medicine, McMaster University, Hamilton, Canada

ⓔ These authors contributed equally to this work.
* meyred@mcmaster.ca

## Abstract

### Background

Little is known about the impact of race/ethnicity on weight change at university. The objective of this study is to determine if ethnicity has an impact on obesity traits in a multiethnic cohort of first-year students at McMaster University in Ontario, Canada.

### Methods

183 first year students from the three most represented ethnic groups (South Asian, East Asian, and white-Caucasian) in our study sample were followed longitudinally with data collected early in the academic year and towards the end of the year. Obesity parameters including body weight, body mass index (BMI), waist and hip circumference, and waist hip ratio (WHR) were analyzed. The Wilcoxon signed-rank test was used for pairwise comparison of traits from the beginning to the end of the year in the absence of adjustments. Linear regression was used with covariate adjustments to investigate the effect of ethnicity on obesity traits.

### Results

A significant increase in weight by 1.49 kg (95%CI: 1.13–1.85) was observed over the academic year in the overall analyzed sample. This was accompanied by significant gains in BMI, waist and hip circumferences, and WHR. Ethnicity stratified analysis indicated significant increase in all investigated obesity traits in East Asians and all traits, but WHR, in South Asians. White-Caucasians only displayed significant increases in weight and BMI. Body weight and hip circumference were significantly lower in East Asians compared to white-Caucasians at baseline. However, East Asians displayed a significantly larger increase in mean BMI and weight compared to white-Caucasians after first-year. South Asians displayed larger waist circumference at baseline compared to East Asians and larger WHR compared to white-Caucasians.

**Data Availability Statement:** The data underlying the results presented in the study are available as a Supporting information file.

**Funding:** DM holds a Canada Research Chair in Genetics of Obesity. TS is supported by the Canadian Institutes of Health Research Canada Graduate Scholarship. The funders had no role in study design, data collection and analysis, decision to publish, or preparation of the manuscript.

**Competing interests:** The authors have declared that no competing interests exist.

## Conclusion

Our findings demonstrate that ethnicity has an impact on obesity traits in first-year university students. Universities should take ethnicity into account while implementing effective obesity prevention programs to promote healthy and active lifestyles for students.

## Introduction

Obesity is defined by the World Health Organization (WHO) as abnormal or excessive fat accumulation that presents a risk to health. WHO classifies adults with a body mass index (BMI) over 30 kg/m$^2$ as being obese. Obesity is a major global health concern that affects 650 million adults and is projected to rise to 1.12 billion by 2030 [1,2]. High-income countries such as Canada experience a higher prevalence of obesity [1]. Approximately 63.1% of Canadians were either overweight or obese in 2018, according to the Canadian Community Health Survey (CCHS). A considerable increase from approximately 23.24% to 31.2% was also noted amongst Canadian individuals aged 18–19 between 2010 and 2018. While education status is negatively correlated with BMI in the general population from high-income countries, young adults with higher education gain more body weight (BW) and are more likely to be obese than those without university education in the United States [3–5]. The "Freshman 15" concept suggests that university undergraduate students gain 15 pounds (6.8 kg) during their first year of post-secondary education, although the average weight gain reported in literature is estimated to be 3–5 pounds (1.4–2.3 kg) [6–8]. The shift from adolescence to adulthood is a critical time period for establishing healthy behaviours and is associated with risk of chronic disorders [9]. Particularly, students who pursue post-secondary education are at increased risk of weight gain than those who do not due to environmental stimuli and psychosocial factors [10,11]. Obesity, especially if developed early in life, is associated with the rapid onset of multiple comorbidities (e.g. depression, sleep disorders, osteoarthritis, dyslipidemia, type 2 diabetes, hypertension, cardiovascular disease, cancers), lower quality of life and premature mortality [9,12]. Treatments such as behavioral and lifestyle interventions, therapeutics, and bariatric surgery exist; however, despite the investment of significant resources in developed countries such as Canada, obesity is difficult to reverse and tends to be a chronic disorder [13,14]. In that context, researching the causes of obesity in young adults is a critical step to improve the prediction, prevention and treatment of obesity in future generations [15,16].

The modern obesity epidemic is explained by major environmental changes such as an unhealthy diet and physical inactivity among many other factors [17]. However, not everybody exposed to an 'obesogenic' environment becomes obese because of inter-individual biological differences (ie. *in utero* programming, age, sex, gut microbiome, epigenetics and genetics) [16]. Race/ethnicity (hereafter referred to as ethnicity) is a determinant of obesity at the interface of biology and environment [18]. It is defined as a group of people with similar cultural and biological characteristics [18]. Ethnicity has been associated with a differential risk of obesity in diverse multiethnic countries, including Canada [19]. Few studies have focused on the impact of ethnicity on obesity traits in undergraduate students during their freshman year [20–22]. As no data on this topic are available in young adults from Canada, we investigated the impact of ethnicity on the change in obesity traits during the freshman year in a multiethnic prospective cohort of 183 undergraduate students at McMaster University.

## Participants and methods

### Participants

Genetic and EnviroNmental Effects on weight in University Students (GENEiUS) was a prospective observational study which investigated the environmental and biological determinants of obesity trait changes in Canadian undergraduate students [15]. Undergraduate students from McMaster University (Hamilton, Ontario) were followed every six months over four years beginning in September of their first year of study. First year students enrolled at McMaster University, between the ages of 17 and 25, were eligible to participate in the study. Participants were primarily recruited via in-class advertising on main university campus and through social media promotion. Particularly, a non-probabilistic sampling approach was utilized whereby anyone who showed interest and met the eligibility criteria was enrolled into the study. Individuals who were pregnant, had previously given birth, or had a medical condition that could have impacted BMI for a long period of time (e.g. bariatric surgery, immobilization from injury) were excluded from the study. Additional details regarding the GENEiUS study have been described previously [15]. Written informed consent was obtained directly from all participants. All methods and procedures for this study were in accordance with the Declaration of Helsinki principles and were reviewed and approved by the Hamilton Integrated Research Ethics Board (REB#0524).

### Data collection

Four cohorts of participants (2015–2016, 2016–2017, 2017–2018, 2018–2019) were followed longitudinally with data collected at two study visits: the beginning of their first-year (September/October) and the end of their first-year (March/April). A total of 361 participants were enrolled in the study, of which 245 (68%) completed one year of follow-up. Only 183 participants were analyzed in this investigation (i.e. participants of non-admixed East Asian (N = 76), South Asian (N = 46), and white-Caucasian (N = 61) ethnicities only). A rule of thumb in statistics is that a sample size of at least 30 is sufficiently large to make inferences about the population from the sample [23,24]. Therefore, participants with African, Latin American, Pacific Islander, and Middle-Eastern ethnicities were excluded from the analysis in the present investigation because their sample sizes were insufficient. Data analyzed in this study included anthropometrics (body weight (BW), BMI, waist circumference (WC), hip circumference (HC), waist hip ratio (WHR), and demographics (sex, age, ethnicity, living status, type of undergraduate program).

### Phenotypes

The obesity trait outcome variables including BW, BMI, WC, HC, and WHR were examined. Trained research personnel performed all anthropometric measurements in duplicate to reduce intra-rater variability. Participants wore light clothing and removed shoes before being weighed. BW was measured to the nearest 0.1 kg using a digital scale (Seca, Hamburg, Germany). Height was measured to the nearest 0.1 cm using a portable stadiometer (Seca 225, Hamburg, Germany). WC was measured after a normal exhalation at the midpoint of the last palpable rib and the superior portion of the iliac crest to the nearest 0.1 cm and HC was measured at the widest part of the buttocks to the nearest 0.1 cm using a stretch-resistant tape measure, as previously described by the World Health Organization (WHO) [25]. WHR was calculated as WC divided by HC. BMI ($kg/m^2$) was calculated by dividing weight by squared height. Demographic information (sex, age, ethnicity, living status, and type of undergraduate program) was collected through online, self-reported questionnaires.

## Statistical methods

All statistical analyses were performed using IBM SPSS Version 25 statistical package. Descriptive analysis was carried out to assess the baseline distribution of traits within the study sample. Data for continuous variables have been reported using means and standard deviations while categorical data have been reported by counts and percentages. Anthropometric data at each time point were screened for potential outliers. Any identified outlying data points were individually cross-checked to determine if they were true outliers, representing participants who truly fell outside the general distribution of our data, or if the outliers were a result of inaccuracies in measurement or data transcription. Data inaccuracies were corrected while all other outliers were left in the dataset. All data were assessed graphically and statistically for normality of distribution prior to analysis. The non-parametric Wilcoxon signed-rank test was used for pairwise comparisons of obesity traits (i.e. BW, BMI, WC, HC, WHR) at baseline and after 6 months (i.e. beginning and end of the 1st year). The effect of ethnicity on obesity traits at baseline and on the change of parameters by the end of first year were tested using linear regression models with adjustment for covariates including sex, cohort of recruitment (i.e. 2015–2016, 2016–2017, 2017–2018, 2018–2019), and baseline trait values. A rank-based inverse normal transformation was applied in cases where the assumption of normality was violated. In this case, based on the fact that i) the present study is hypothesis-driven; ii) the research questions have been previously tested in literature; iii) the tested obesity outcomes are not independent, a Bonferroni correction was not applied as even though it reduces the chance of making type I errors, it increases the chance of making type II errors [26,27]. Therefore, the level of statistical significance was set at p <0.05 for all tests.

## Results

### Participant characteristics

Of the 361 participants enrolled in the study, 245 (68%) completed one year of follow up. Only 183 participants were analyzed in this investigation (i.e. participants of East Asian, South Asian, and white-Caucasian ethnicities only). East Asians represented 41.5% of the sample (n = 76), white-Caucasian represented 33.3% (n = 61), and South Asians represented 25.1% (n = 46). Participants with African, Latin American, Pacific Islander, and Middle-Eastern ethnicities were excluded from the study, because their sample sizes were insufficient (n ≤ 20 for all). Participants displayed an average age of 17.84(SD = 0.49) years at baseline. Male and female participants represented 18.6% (n = 34) and 81.4% (n = 149) of the sample, respectively. 74.3% of the analyzed sample lived in residence on campus (n = 136), 14.8% lived at home with family (n = 27), 10.4% lived in a student house off campus (n = 19) and 0.5% did not report living arrangement status (n = 1). Among those who reported their program of study, 89.1% (n = 155) reported being enrolled in a science based academic program (e.g. Health Science, Life Science, Kinesiology, Engineering) while 10.9% (n = 19) reported being in enrolled a non-science academic program (e.g. Humanities, Business, Arts). At the beginning of the year, 79.2% (n = 145) of the participants had a normal BMI, 12.6% were underweight (n = 23), 7.1% were overweight (n = 13), and 1.1% (n = 2) were obese.

### Overall changes in obesity traits in first year of university

Table 1 summarizes the changes in obesity traits during the first year of university in 183 participants. A statistically significant increase across the five investigated obesity parameters was noted between the two time points. The average body weight increased from 59.44 (SD = 10.04) kg to 60.93 (SD = 10.31) kg over the year, corresponding to a gain of 1.49 kg (3.28

**Table 1. Overall trends in first year of university.**

|  | Beginning Mean (SD) | End Mean (SD) | Change MD (95% CI) | P-value* |
|---|---|---|---|---|
| **Body Weight (kg)** | 59.44 (10.04) | 60.93 (10.31) | 1.49 (1.13–1.85) | **<0.001** |
| **BMI (kg/m²)** | 21.32 (2.71) | 21.91 (2.77) | 0.59 (0.46–0.73) | **<0.001** |
| **Waist Circumference (cm)** | 74.59 (7.62) | 75.93 (8.01) | 1.34 (0.73–1.95) | **<0.001** |
| **Hip Circumference (cm)** | 96.54 (6.42) | 97.42 (5.99) | 0.88 (0.42–1.33) | **<0.001** |
| **WHR** | 0.772 (0.049) | 0.779 (0.054) | 0.007 (0.001–0.012) | **0.017** |

Data are expressed as mean (SD) and mean difference (95% CI); Abbreviations: BMI, body mass index; WHR, Waist to hip ratio; MD, Mean difference.

*Non-parametric pairwise comparison (non-adjusted comparison of change in outcomes from beginning to end of school year).

P-values below 0.05 represented in bold font.

pounds). Overall, 60.7% of the analyzed sample (n = 111) gained more than 1 kg, 14.2% (n = 26) lost more than 1 kg, and 25.1% remained within 1kg of their baseline weight. An increase in average BMI from 21.32 (SD = 2.71) kg/m² to 21.91 (SD = 2.77) kg/m² was also observed, corresponding to a change of 0.59 kg/m² (95% CI: 0.46–0.73). It is important to note that the average BMI at both time points was below 25 kg/m². It signifies that a majority of participants remained within the 'normal weight' category from the beginning to the end of the year. Increases in WC and HC, by 1.34 cm (95% CI: 0.73–1.95) and 0.88 cm (95% CI: 0.42–1.33) respectively, were also observed. An increase in WHR from 0.772 (SD = 0.049) to 0.779 (SD = 0.054) was also noted between the two time points.

## Impact of ethnicity on obesity traits in first year of university

Table 2 summarizes the changes in anthropometric traits during the first year of University in the three ethnic groups. A statistically significant increase across a majority of the obesity parameters was noted between the two time points in the East Asian, white-Caucasian, and South Asian ethnic groups. A notable exception was the absence of a significant increase in the WC and HC of white-Caucasians. Similarly, while East Asians displayed a significant increase in WHR from 0.769 ± 0.045 to 0.781 ± 0.048, no such trends were observed in South Asians and white-Caucasians.

Table 3 compares the differences in obesity traits at baseline and change over first year of university between the three analyzed ethnic groups. Body weight was significantly lower in East Asians (57.37 kg, SD = 9.85) and South Asians (60.19 kg, SD = 11.37) than in white-Caucasians (61.47 kg, SD = 8.81) at baseline after adjustment for sex and cohort of recruitment (p < 0.05 for both comparisons, refer to Table 3). Similarly, hip circumference was significantly lower in East Asians (94.97 cm, SD = 5.95) compared to white-Caucasians (97.93 cm, SD = 5.49) at baseline (P = 0.007). On the contrary, when examining change, East Asians displayed significantly larger increases over the academic year than white-Caucasians in BW [1.82 kg (95% CI: 1.35–2.30) versus 0.94 kg (95% CI: 0.39–1.48); P = 0.026] and BMI [0.79 kg/m² (95% CI: 0.60–0.99) versus 0.40 kg/m² (95% CI: 0.20–0.60); P = 0.012]. At baseline, waist circumference was significantly higher in South Asians (77.01 cm, SD = 9.50) compared to East Asians (73.05 cm, SD = 6.80; P = 0.039), and WHR was significantly higher in South Asians (0.790, SD = 0.052) than white-Caucasians (0.762, SD = 0.049, P = 0.020).

## Discussion

The cohort under investigation is multi-ethnic, consisting of 7 ethnic groups, and reflects the general population of Ontario. However, the over-representation of South Asians and East

**Table 2. Ethnicity specific trends by East Asian (n = 76), white-Caucasian (n = 61), and South Asian (n = 46) subgroups.**

| Anthropometric Trait | Ethnicity | Beginning Mean (SD) | End Mean (SD) | Change MD (95% CI) | P-value* |
|---|---|---|---|---|---|
| **Body Weight (kg)** | East Asian | 57.37 (9.85) | 59.19 (9.67) | 1.82 (1.35–2.30) | **<0.001** |
| | Caucasian | 61.47 (8.81) | 62.41 (9.20) | 0.94 (0.39–1.48) | **0.002** |
| | South Asian | 60.19 (11.37) | 61.85 (12.36) | 1.67 (0.69–2.64) | **0.001** |
| **BMI (kg/m$^2$)** | East Asian | 21.05 (2.45) | 21.85 (2.40) | 0.79 (0.60–0.99) | **<0.001** |
| | Caucasian | 21.34 (2.17) | 21.74 (2.29) | 0.40 (0.20–0.60) | **<0.001** |
| | South Asian | 21.72 (3.62) | 22.25 (3.77) | 0.53 (0.18–0.88) | **0.003** |
| **Waist Circumference (cm)** | East Asian | 73.05 (6.80) | 74.90 (6.82) | 1.84 (0.99–2.70) | **<0.001** |
| | Caucasian | 74.66 (6.55) | 75.32 (6.75) | 0.65 (-0.22–1.53) | 0.233 |
| | South Asian | 77.01 (9.50) | 78.44 (10.59) | 1.43 (-0.21–3.07) | **0.050** |
| **Hip Circumference (cm)** | East Asian | 94.97 (5.95) | 95.88 (5.36) | 0.91 (0.22–1.60) | **0.014** |
| | Caucasian | 97.93 (5.49) | 98.60 (4.93) | 0.67 (-0.13–1.47) | 0.135 |
| | South Asian | 97.29 (7.75) | 98.39 (7.61) | 1.10 (0.12–2.09) | **0.036** |
| **WHR** | East Asian | 0.769 (0.045) | 0.781 (0.048) | 0.012 (0.003–0.021) | **0.004** |
| | Caucasian | 0.762 (0.049) | 0.763(0.050) | 0.001 (-0.008–0.011) | 0.866 |
| | South Asian | 0.790 (0.052) | 0.795 (0.062) | 0.005(-0.008–0.017) | 0.481 |

Data are expressed as mean (SD) and mean difference (95% CI); Abbreviations: BMI, body mass index; WHR, Waist to hip ratio; MD, Mean difference.

*Non-parametric pairwise comparison by ethnicity subgroup (non-adjusted comparison of change in outcomes from beginning to end of school year). P-values below 0.05 represented in bold font.

† The East Asian ethnicity describes individuals whose ancestors were originally from East Asian countries such as China, Japan, North and South Korea, Vietnam, and Philippines. The South Asian ethnicity describes individuals whose ancestors were originally from South Asian countries such as India, Bangladesh, Pakistan, and Sri-Lanka. White-Caucasian ethnicity describes individuals with European ancestry.

Asians in the sample may not reflect the percentages currently observed in the general population of Ontario. According to the 2016 Canadian Census by Statistics Canada, 8.9% and 10.7% of Ontario's population are South Asians and East Asians respectively [28]. In comparison, the proportion of South Asians and East Asians in our study sample was 25.1% and 41.5%

**Table 3. Association between ethnicity and obesity traits in first year of university.**

| | | East Asian vs. Caucasian β (Std. Error) and p-value | South Asian vs. Caucasian β (Std. Error) and p-value | South Asian vs. East Asian β (Std. Error) and p-value |
|---|---|---|---|---|
| **Body Weight (kg)** | Baseline[1] | -0.460 (0.152); **0.003** | -0.404 (0.174); **0.021** | 0.056 (0.168); 0.737 |
| | Change[2] | 0.399 (0.178); **0.026** | 0.300 (0.202); 0.140 | -0.100 (0.193); 0.607 |
| **BMI (kg/m$^2$)** | Baseline[1] | -0.107 (0.174); 0.541 | -0.033 (0.200); 0.870 | 0.074 (0.193); 0.702 |
| | Change[2] | 0.433 (0.170); **0.012** | 0.157 (0.195); 0.421 | -0.275 (0.188); 0.145 |
| **Waist Circumference (cm)** | Baseline[1] | -0.278 (0.162); 0.089 | 0.095 (0.186); 0.610 | 0.373 (0.180); **0.039** |
| | Change[2] | 0.315 (0.172); 0.069 | 0.225 (0.196); 0.254 | -0.090 (0.192); 0.640 |
| **Hip Circumference (cm)** | Baseline[1] | -0.460 (0.168); **0.007** | -0.198 (0.192); 0.305 | 0.262 (0.186); 0.160 |
| | Change[2] | -0.110 (0.164); 0.503 | 0.019 (0.185); 0.918 | 0.129 (0.179); 0.470 |
| **WHR** | Baseline[1] | 0.092 (0.160); 0.564 | 0.429 (0.183); **0.020** | 0.337 (0.177); 0.059 |
| | Change[3] | 0.291 (0.154); 0.061 | 0.207 (0.177); 0.244 | -0.084 (0.171); 0.625 |

[1]Linear regression with inverse normal rank transformation, adjusted for sex and cohort;

[2]Linear regression with inverse normal rank transformation, adjusted for sex, cohort and baseline values;

[3]Linear regression with inverse normal rank transformation, adjusted for sex, cohort, baseline WHR, baseline BMI, and BMI Change; Abbreviations: BMI, body mass index; WHR, Waist to hip ratio.

respectively. The overrepresentation of students from these two ethnic groups can be explained by family structure, household income and parental education among other factors. Previous studies have shown that educational attainment is higher in children whose parents have higher education [29]. In addition, in their report, Krahn & Taylor (2005) have also noted that first and second generation Canadian students that make up 52.4% of Ontario's population are more likely to pursue post-secondary education due to the "immigrant drive" that suggests higher parental expectations [29]. Similarly, a national survey in Canada of prospective students and their parents show that students from visible minorities also have higher educational aspirations [30].

At baseline, the sample had an average weight of 59.44 kg and an average BMI of 21.32 kg/m$^2$, which is within the normal range of 18.50–24.99 according to WHO guidelines [31]. By the end of the year, participants displayed significant weight gain of 1.49 kg or 3.28 pounds. This is in accordance with previous meta-analyses estimates of 1.36 kg and 1.75 kg [7,8]. Beaudry et al (2019) also found that males and females gained 3.8 kg and 1.8 kg respectively at Brock University in Ontario [32]. This suggests that universities may be an "obesogenic" environment for young adults across Ontario as well as in other provinces across Canada. The Freshman 15 phenomenon, better known as the Freshman 5 phenomenon, can be explained by increased stress, increased alcohol and fast-food consumption, decreased physical activity and lack of sleep [7,10,33].

South Asians displayed a significantly higher WHR at baseline compared to white-Caucasians. This is consistent with previous studies that have established that South Asian adults have a higher propensity to gain abdominal and visceral fat than their white-Caucasian counterparts [34–36]. This is particularly concerning because accumulation of fat in the abdominal area is associated with an increased risk for cardio-metabolic diseases such as type 2 diabetes and coronary heart disease [37]. Our findings are also compatible with the observation that South Asians have increased abdominal visceral fat despite having a healthy BMI, and demonstrate that abdominal fat deposition in South Asians starts early in life likely due to biological factors (e.g. *in utero* environment, genetics, epigenetics). In comparison, East Asians in our sample displayed lower BW, WC and HC at inclusion. These lower values at baseline can be also potentially be attributed to biological (e.g. genetics, epigenetics, microbiome) and environmental factors. Particularly, East Asian diet patterns have been found to be associated with a lower risk of abdominal obesity. Their diets are characterized by a high intake of whole grains and vegetables, thus a higher intake of fibre, and a decreased risk of obesity [38]. It should be noted that although East Asians have been generally shown to have a lower BMI, they have been reported to have a higher percentage of body fat compared to white-Caucasians, similar to South Asians [39]. Interestingly, when examining change over the academic year, East Asians displayed larger increases in most traits compared to the other ethnic groups. This suggests that they are particularly at risk of weight gain and unhealthy fat deposition during the transition to university. The significant weight gain observed in this case may be attributed to the change in diet, increased sedentary behaviours, increased stress, and living away from home accompanied by less parental supervision. Their traditional diet does not necessarily continue on campus. Instead, a Westernized diet characterized by high intake of fat and low intake of fibre is undertaken as a result of the campus environment [38]. It is important to note here that the observed weight gain may not be solely attributed to an increase in fat, but also other potential factors such as overall growth or a gain in muscle mass. When examining the average height in our sample, we did not observe an increase from baseline (166.7 cm, SD = 8.17) to the end of the year (166.5 cm, SD = 8.11). Unfortunately, in this case, we did not directly measure or calculate body fat percentage and hence could not account for that variable. Nevertheless, based on our examination of adiposity indicators such as waist and hip

circumference, we saw that East Asian participants displayed significant increases in those areas over the academic year. Hence ultimately, based on our data, we postulate that the observed weight gain in our sample may be attributed to a mix of fat and muscle increase.

One of the strengths of this study is that there are not very many studies on this particular topic as the effect of ethnicity on obesity traits has never been investigated in this population group in Ontario. The study is also longitudinal in nature allowing for stronger inferences regarding the predictors of incident weight change in first year of university. Moreover, we further investigated change in various obesity parameters such as WHR, WC and HC to assess adiposity. Finally, the study was conducted in Ontario, a highly diverse population, which is ideal for investigating the effect of different ethnicities on obesity traits.

There are some limitations of the GENEiUS Study that should be noted. The study has a modest sample size, therefore, particular ethnic groups were not accounted for. While applying a Bonferroni correction for multiple tests can be over-conservative, using a threshold for significance at $P < 0.05$ can be too liberal. Therefore, we strongly recommend to replicate our more promising findings in additional studies. Our study also exhibits gender bias as more than three fourths of the sample consists of women; however, this trend is commonly seen in epidemiological studies. Apart from that, participant attrition in this case may have potentially biased our conclusions. Lastly, it is important to note that ethnicity was self-reported.

In conclusion, all three ethnic groups experienced significant weight gain. A significant increase in all five obesity traits was only observed in East Asians despite their low baseline values relative to the other ethnic groups. This indicates an increased risk for unhealthy fat deposition in response to an obesity-prone environment. At baseline, South Asians started with a relatively high BMI and ended with a relatively high BMI, however, the increase was less compared to East Asians. White-Caucasians maintained a BMI higher than East Asians and lower than South Asians, and experienced the least change in all five obesity traits at the end of the first year of university. Our research can help design effective interventions with ethnic-specific guidelines, especially as individuals of different ethnicities may appear to be healthy with normal BMI levels, but have a higher than normal percentage of body fat. Understanding how ethnicity impacts body weight changes in young adults is critical to combat the rise of adult obesity.

## Supporting information

**S1 Dataset.**
(XLSX)

## Acknowledgments

We are indebted to all participants of this study. We would also like to extend our thanks to Anika Shah, Roshan Ahmad, Adrian Santhakumar, Kelly Zhu, Guneet Sandhu, Dea Sulaj, Tina Khordehi, Ansha Suleman, Heba Shahaed, Andrew Ng, Tania Mani, Sriyathavan Srichandramohan, Deven Deonarain, Celine Keomany, Omaike Sikder, Isis Lunsky, Gurudutt Kamath, and Christy Yu for their help with data collection. DM holds a Canada Research Chair in Genetics of Obesity. TS is supported by the Canadian Institutes of Health Research Canada Graduate Scholarship.

## Author Contributions

**Conceptualization:** David Meyre.

**Data curation:** Tanmay Sharma.

**Formal analysis:** Tanmay Sharma, David Meyre.

**Funding acquisition:** David Meyre.

**Investigation:** Tanmay Sharma, Baanu Manoharan, Christine Langlois, Rita E. Morassut, David Meyre.

**Methodology:** Tanmay Sharma, David Meyre.

**Project administration:** David Meyre.

**Resources:** David Meyre.

**Supervision:** David Meyre.

**Validation:** David Meyre.

**Writing – original draft:** Tanmay Sharma, Baanu Manoharan, David Meyre.

**Writing – review & editing:** Christine Langlois, Rita E. Morassut.

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
