## [Decision Letter · Decision Letter 0]

11 Sep 2020

PONE-D-20-20880

The effect of race/ethnicity on obesity traits in first year university students from Canada: the GENEiUS study

PLOS ONE

Dear Dr. Meyre,

Thank you for submitting your manuscript to PLOS ONE. After careful consideration, we feel that it has merit but does not fully meet PLOS ONE’s publication criteria as it currently stands. Therefore, we invite you to submit a revised version of the manuscript that addresses the points raised during the review process.

We look forward to receiving your revised manuscript.

Kind regards,

Oathokwa Nkomazana, MD MSC PhD

Academic Editor

PLOS ONE

Additional Editor Comments:

Thank you for the interesting research. Please note and attend to the reviewers' comments, paying particular attention to the concerns of Reviewer 1.

Journal Requirements:

Reviewers' comments:

Reviewer's Responses to Questions

**Comments to the Author**

1. Is the manuscript technically sound, and do the data support the conclusions?

Reviewer #1: Partly

Reviewer #2: Yes

2. Has the statistical analysis been performed appropriately and rigorously? 

Reviewer #1: Yes

Reviewer #2: Yes

3. Have the authors made all data underlying the findings in their manuscript fully available?

Reviewer #1: No

Reviewer #2: Yes

4. Is the manuscript presented in an intelligible fashion and written in standard English?

Reviewer #1: Yes

Reviewer #2: Yes

5. Review Comments to the Author

Reviewer #1: 0Lines 92-103 Report in the past tense e.g. line 97, ... between the ages of 17 and 25 "were" eligible ad not "are" eligible

Lines 95 After advertising and use of inclusion criteria, how were students finally selected to particiate in the study (sampling)

Lines 113-115 Be clear whether the mentioned particpants were excluded from the study or from the data analysis. This is because your exclusion criteria does not include race/ethnicity

Lines 115-117 refers to data analysis in a section discussing data collection. Move to the data analysis section.

Lines 143-144 The non parametric pairwise test used is not fully explained

Lines 52-153 The justification for leaving the p value at 0.05 after applying the Bonferroni correction is not clear. Why is the P=0.05 not divided by the number of tests?

Line 205 Table 3 should be Table 2. This is because all the information refers to table 2 and not 3

Lines 210-212 This sentence refers to change in BMI whereas the focus is on baseline status

After 216 Missing - results reportig for Table 3.

After 236 There description of the gender variable (female versus male) is missing

Line 264 - How was percentage body fat measured or calculated?

Missing -discussions on the reslts of Table Association between ethinicity and obesity traits in first year at university

Reviewer #2: Overall, a simple and straightforward study of the relationship between ethnicity and body composition changes among first year students at a university.

All the body composition changes were statistically significant when baseline and 6 months values were compared. I am not certain these changes are clinically important, particularly as illustrated by BMI. We are fortunate as we have a normal range for BMI which spans about 5 units. The change is within the normal range.

The average age at enrollment into the study was approximately 18 years and the majority were women. I would like the authors to mention the adolescent growth spurt and peak either in discussion or introduction. A comparison of height measures at baseline and 6 months would be useful to confirm/illustrate the issue of peak height. In the discussion, the weight gain has not been dissociated from any possible training effects which could have increased muscle mass. Even after peak height has been achieved, a great muscle mass could be gained and could influence weight. Would like that to taken on board in the discussion.

As I was reading, I got the impression that there was an assumption that the use of the phrases East and South Asians would be understood by all the readers. A footnote or short definition in the methods section or at the bottom of Table 2 is suggested.

Did any student in the cohort lose weight during the study period? it would be interesting to know how many did.

Line 88, replace "subjects' with "participants"

In the discussion, line 229, a reference is required for Statistics Canada

Line 233, a reference number [29] should be added at the end of the sentence or rename [28] to maintain order of references.

Line 239, what is the meaning of " relatively normal BMI"?

Line 240, is it meta-analyses or meta-analysis? either way reference or references would be required at the end of that sentence.

Line 243, replace 'confirms' with a hedging word such as "suggests'?

Line 265, "weight gain is", replace "is' with a hedging word. In this particular study, data on diet, stress levels etc has not be reported.

Under Acknowledgements: what does "technical assistance" mean? I would be happier to read a paper that acknowledges 'research assistants or data collectors' as having helped with data collection. Recommendation is to replace 'technical assistance" with "help with data collection".

Under references, ref 25 requires editing ie Organization, W.H? probably WHO?

6. PLOS authors have the option to publish the peer review history of their article (what does this mean?). If published, this will include your full peer review and any attached files.

Reviewer #1: No

Reviewer #2: No

---

## [Author Response · Author response to Decision Letter 0]

17 Oct 2020

We would like to thank the editor and the reviewers for their exceptional input and suggestions on the article. We have addressed their comments to the best of our ability and we think that the revised version of the manuscript has significantly improved. 

Reviewer #1: 

Lines 92-103 Report in the past tense e.g. line 97, ... between the ages of 17 and 25 "were" eligible ad not "are" eligible

We thank the reviewer for this comment. We have now changed those lines to past tense. We also re-reviewed the entire manuscript to correct any other grammatical or reporting errors that may have been present.

Lines 95 After advertising and use of inclusion criteria, how were students finally selected to particiate in the study (sampling)

We thank the reviewer for this comment. We have now included an additional sentence to clarify this.

“Particularly, a non-probabilistic sampling approach was utilized whereby anyone who showed interest in participating and met the eligibility criteria was enrolled into the study.”

Lines 113-115 Be clear whether the mentioned participants were excluded from the study or from the data analysis. This is because your exclusion criteria does not include race/ethnicity

The reviewer is right. It has now been specified that the mentioned participants were particularly excluded from the data analysis.

Lines 115-117 refers to data analysis in a section discussing data collection. Move to the data analysis section.

The reviewer brings up a valid point. However, while that particular sentence presents an analytical consideration, we feel that it fits well in that section as it provides rationale to the reader for why we particularly chose to examine the groups that we did. While we can add that line under the statistical methods section of our paper, we feel that it is an important point that may get missed among the other analytical details included in that section. 

Lines 143-144 The non parametric pairwise test used is not fully explained

We thank the reviewer for the comment. The particular non-parametric tests used have now been specified.

“The non-parametric tests Wilcoxon signed-rank test, were used for pairwise comparisons of obesity traits (i.e. BW, BMI, WC, HC, WHR) at baseline and after 6 months (i.e. beginning and end of the 1st year).”

Lines 52-153 The justification for leaving the p value at 0.05 after applying the Bonferroni correction is not clear. Why is the P=0.05 not divided by the number of tests?

In our statistical methods section, we included a sentence with three reasons as to why we did not apply the Bonferroni correction as indicated below. We have slightly modified the wording to improve clarity.

“In this case, based on the fact that i) the present study is hypothesis-driven; ii) the research questions have been previously tested in literature; iii) the tested obesity outcomes are not independent, a Bonferroni correction was not applied as even though it reduces the chance of making type I errors, it increases the chance of making type II errors [26,27]. Therefore, the level of statistical significance was set at p <0.05 for all tests.”

However, we understand the reviewer’s concern, and we have added a sentence in the limitation section to reflect his excellent comment:

“While applying a Bonferroni correction for multiple tests can be over-conservative, using a threshold for significance at P < 0.05 can be too liberal. Therefore, we strongly encourage to replicate our more promising findings in additional studies.”

Line 205 Table 3 should be Table 2. This is because all the information refers to table 2 and not 3

While that paragraph references certain values from Table 2 to illustrate certain points, the core results that form the basis of that paragraph are from Table 3 (i.e. results from the regression analysis evaluating the impact ethnicity on obesity traits at baseline and change). The content of that paragraph highlights the comparison across the different ethnic groups and particularly provides the relevant p-values from the regression analysis, which are presented in Table 3. The paragraph has been modified with more clear references to the results from the cross-ethnicity comparison presented in Table 3.

“Table 3 compares the differences in obesity traits at baseline and change over first year of university between the three analyzed ethnic groups. Body weight was significantly lower in East Asians (57.37 kg, SD = 9.85) and South Asians (60.19 kg, SD = 11.37) than in white-Caucasians (61.47 kg, SD = 8.81) at baseline after adjustment for sex and cohort of recruitment (p < 0.05 for both comparisons, refer to Table 3). Similarly, hip circumference was significantly lower in East Asians (94.97 cm, SD = 5.95) compared to white-Caucasians (97.93 cm, SD = 5.49) at baseline (P = 0.007). On the contrary, East Asians displayed significantly larger increases over the academic year than white-Caucasians in BW [1.82 kg (95% CI: 1.35 – 2.30) versus 0.94 kg (95% CI: 0.39 – 1.48), P = 0.026] and BMI (0.79 ± 0.84 kg/m2 versus 0.40 ± 0.78 kg/m2, P = 0.012). At baseline, waist circumference was significantly higher in South Asians (77.01 cm, SD = 9.50) compared to East Asians (73.05 cm, SD = 6.80; P = 0.039), and WHR was significantly higher in South Asians (0.790, SD = 0.052) than white-Caucasians (0.762, SD = 0.049, P = 0.020).” 

Lines 210-212 This sentence refers to change in BMI whereas the focus is on baseline status

Thank you for the comment. We have updated that sentence to specify that it is referring to change.

“Table 3 compares the differences in obesity traits at baseline and change over first year of university between the three analyzed ethnic groups. Body weight was significantly lower in East Asians (57.37 kg, SD = 9.85) and South Asians (60.19 kg, SD = 11.37) than in white-Caucasians (61.47 kg, SD = 8.81) at baseline after adjustment for sex and cohort of recruitment (p < 0.05 for both comparisons, refer to Table 3). Similarly, hip circumference was significantly lower in East Asians (94.97 cm, SD = 5.95) compared to white-Caucasians (97.93 cm, SD = 5.49) at baseline (P = 0.007). On the contrary, when examining change, East Asians displayed significantly larger increases over the academic year than white-Caucasians in BW [1.82 kg (95% CI: 1.35 – 2.30) versus 0.94 kg (95% CI: 0.39 – 1.48); P = 0.026] and BMI [0.79 kg/m2 (95% CI: 0.60 – 0.99) versus 0.40 kg/m2 (95% CI: 0.20 – 0.60); P = 0.012]. At baseline, waist circumference was significantly higher in South Asians (77.01 cm, SD = 9.50) compared to East Asians (73.05 cm, SD = 6.80; P = 0.039), and WHR was significantly higher in South Asians (0.790, SD = 0.052) than white-Caucasians (0.762, SD = 0.049, P = 0.020).” 

After 216 Missing - results reportig for Table 3.

Thank you for the comment. As discussed for the comment above, while we did discuss the results of Table 3, we referenced certain values from Table 2 in that paragraph as well to better illustrate and integrate information. Nonetheless, the core results that form the basis of that paragraph are from Table 3. As discussed above, we have now updated that paragraph with more clear references to the results from the cross-ethnicity comparison presented in Table 3.

After 236 There description of the gender variable (female versus male) is missing

Line 236 does not mention gender:

“Additionally, a national survey in Canada of prospective students and their parents show that visible minority students also have higher educational aspirations”.

Therefore, we are unsure about how address the reviewer’s comment. Sorry about that.

Line 264 - How was percentage body fat measured or calculated?

We did not measure or calculate percentage body fat in our study. The line refers to a report that had been cited to illustrate a trend noted in previous literature. We have modified the wording to make that clear.

“It should be noted that although East Asians have been generally shown to have a lower BMI, they have been reported to have a higher percentage of body fat compared to white-Caucasians, similar to South Asians [38].”

Missing -discussions on the results of Table Association between ethnicity and obesity traits in first year at university

Thank you for the comment. This has been addressed in the comments above concerning the discussion for Table 3, which is the association table that the reviewer is referring to.

Reviewer #2: 

1. All the body composition changes were statistically significant when baseline and 6 months values were compared. I am not certain these changes are clinically important, particularly as illustrated by BMI. We are fortunate as we have a normal range for BMI which spans about 5 units. The change is within the normal range.

While we agree with the reviewer that the change in BMI is modest in this case and within the normal range, we believe that the change in body weight noted among this group was considerable. As described in the manuscript, the average weight gain noted among participants was 1.5kg (3.31lbs) with certain ethic groups, such as East Asians, displaying a gain of approximately 1.82 kg (4.01 lbs). 

2. The average age at enrollment into the study was approximately 18 years and the majority were women. I would like the authors to mention the adolescent growth spurt and peak either in discussion or introduction. A comparison of height measures at baseline and 6 months would be useful to confirm/illustrate the issue of peak height. In the discussion, the weight gain has not been dissociated from any possible training effects which could have increased muscle mass. Even after peak height has been achieved, a great muscle mass could be gained and could influence weight. Would like that to taken on board in the discussion.

We thank the reviewer for this comment. We have now added a paragraph in the discussion section of the paper that addresses these points.

“It is important to note here that the observed weight gain may not be solely attributed to gain in fat, but also other potential factors such as overall growth or gain in muscle mass. When examining the average height in our sample, we did not observe an increase from baseline (166.7 cm, SD = 8.17) to the end of the year (166.5 cm, SD = 8.11). Unfortunately, in this case, we did not directly measure or calculate body fat percentage and hence could not account for that variable. Nevertheless, based on our examination of adiposity indicators such as waist and hip circumference, we saw that East Asian participants displayed significant increases in those areas over the year. Hence ultimately, based on our data, we postulate that the observed weight gain in our sample may be attributed a mix of fat and muscle increase.”

3. As I was reading, I got the impression that there was an assumption that the use of the phrases East and South Asians would be understood by all the readers. A footnote or short definition in the methods section or at the bottom of Table 2 is suggested.

Thank you for the comment. As per the reviewer’s suggestion, a description of each ethnic group has been added at the bottom of Table 2. 

4. Did any student in the cohort lose weight during the study period? it would be interesting to know how many did.

The reviewer brings up an interesting point. We have now added the following sentence in the results of the paper, under the section about the ‘overall changes in obesity traits,’ describing the number/proportion of participants who gained weight, lost weight, or maintained their weight based on a 1kg threshold of change. 

 “Overall, 60.7% of the analyzed sample (n = 111) gained more than 1 kg, 14.2% (n = 26) lost more than 1 kg, and 25.1% remained within 1kg of their baseline weight.”

In addition, we have further updated all our tables and texts alike to include the 95% confidence intervals for all values of change, as we believe that a confidence interval is more appropriate in this case and would be more reflective of the spectrum of change observed in the sample for each of the investigated traits. 

5. Line 88, replace "subjects' with "participants"

This has now been changed, thank you. 

6. In the discussion, line 229, a reference is required for Statistics Canada

This has now been changed, thank you. 

7. Line 233, a reference number [29] should be added at the end of the sentence or rename [28] to maintain order of references.

This has now been changed, thank you. 

8. Line 239, what is the meaning of " relatively normal BMI"?

This has now been changed and the word ‘relatively’ has been removed.

Line 240, is it meta-analyses or meta-analysis? either way reference or references would be required at the end of that sentence.

The references have been added. 

10. Line 243, replace 'confirms' with a hedging word such as "suggests'?

Thank you for the comment. This has now been changed.

11. Line 265, "weight gain is", replace "is' with a hedging word. In this particular study, data on diet, stress levels etc has not be reported.

Thank you for the comment. This has now been changed.

“The significant weight gain may be attributable to the change in diet, increased sedentary behaviours, increased stress and living away from home accompanied by less parental supervision.”

12. Under Acknowledgements: what does "technical assistance" mean? I would be happier to read a paper that acknowledges 'research assistants or data collectors' as having helped with data collection. Recommendation is to replace 'technical assistance" with "help with data collection".

This a great suggestion. The change has now been made.

13. Under references, ref 25 requires editing ie Organization, W.H? probably WHO?

The reference has been edited to say “WHO”, good pick.

---

## [Editor Report · Decision Letter 1]

9 Nov 2020

The effect of race/ethnicity on obesity traits in first year university students from Canada: the GENEiUS study

PONE-D-20-20880R1

Dear Dr. Meyre,

We’re pleased to inform you that your manuscript has been judged scientifically suitable for publication and will be formally accepted for publication once it meets all outstanding technical requirements.

Kind regards,

Oathokwa Nkomazana, MD MSC PhD

Academic Editor

PLOS ONE
---

## [Editor Report · Acceptance letter]

13 Nov 2020

PONE-D-20-20880R1 

The effect of race/ethnicity on obesity traits in first year university students from Canada: the GENEiUS study 

Dear Dr. Meyre:

I'm pleased to inform you that your manuscript has been deemed suitable for publication in PLOS ONE. Congratulations! Your manuscript is now with our production department. 

Kind regards, 

on behalf of

Dr. Oathokwa Nkomazana 

Academic Editor

PLOS ONE